# Seroepidemiology of hepatitis A, B, C, D and E virus infections in the general population of Peru: A cross-sectional study

**Cesar Cabezas[1,5]\*, Omar Trujillo[2], Ángel Gonzales-Vivanco[3], Carlos Manuel Benites Villafane[3], Johanna Balbuena[1], Alfredo Oswaldo Borda-Olivas[4], Magna Aurora Suarez-Jara[1], Flor de María Peceros[1], Max Carlos Ramírez-Soto[1]**

**1** Centro Nacional de Salud Pública, Instituto Nacional de Salud, Lima, Peru, **2** Centro Nacional de Salud Intercultural, Instituto Nacional de Salud, Lima, Peru, **3** Estrategia Sanitaria de Prevención y Control de Infecciones de Transmisión Sexual, VIH/SIDA y Hepatitis B, Ministerio de Salud, Lima, Peru, **4** Instituto Nacional de Salud del Niño–San Borja, Lima, Peru, **5** Facultad de Medicina, Universidad Nacional Mayor de San Marcos, Lima, Peru

\* ccabezas@ins.gob.pe

**Data Availability Statement:** Instituto Nacional de Salud research data cannot be publicly shared at this time. The data are also not available in an institutional repository, since contain potentially

## Abstract

### Background

Viral hepatitis (hepatitis A, B, C, D and E) remains a public health problem in Peru, with a high disease burden. There are limited data on the prevalence of viral hepatitis at a national level, and none reported for over two decades. In this study, the prevalence rates of hepatitis A (HAV), B (HBV), C (HCV), D (HDV) and E virus (HEV) infections in the Peruvian population were determined to provide updated baseline data that would help guide the development of strategies aimed at reducing the transmission of viral hepatitis in Peru.

### Methods

We conducted a cross-sectional, population-based study in the 25 regions of Peru. The study included participants of both sexes, aged 15–69 years, who had lived for >6 months in a specific region of Peru. Serum samples were analyzed by ELISA for anti-HAV (IgG), anti-HBs ≥10 mUI/ml, anti-HCV, anti-HDV and anti-HEV (IgG) antibodies, and by chemiluminescence for the HBV surface antigen (HBsAg) and antibodies against the core HBV antigen (anti-HBc IgM and IgG).

### Results

In a total of 5183 study participants, the prevalence rates of anti-HAV (IgG), HBsAg, total anti-HBc IgG, anti-HBs ≥10 mUI/ml, anti-HCV and anti-HEV (IgG) were 98.4% [95% confidence interval (CI) 98.0–98.7), 0.4% (95% CI 0.21–0.55), 10.1% (95% CI 9.4–11.0), 60% (95% CI 58.5–61.2), 0.1% (95% CI 0.02–0.25), and 14% (95% CI 13.1–15.0%), respectively. The prevalence of anti-HDV among HBsAg carriers was 15% (3/20).

sensitive patient information. All interested researchers can access these data through Instituto Nacional de Salud, Peru (Cápac Yupanqui 1400 - Jesus María, Lima 11, Perú; Teléfono: 511 748 1111) subject to review by the Direction, ethics approval, or signing of a data sharing agreement. Data are provided only once a data sharing agreement is in place between Instituto Nacional de Salud (the custodian of the data) and the researchers or an institution. For more information about data access, see https://web.ins.gob.pe/. Applications are submitted to comunicaciones@ins.gob.pe.

**Funding:** This study was funded by the Instituto Nacional de Salud, Peru (project OI-082-13).

**Competing interests:** The authors have declared that no competing interests exist.

## Conclusions

The prevalence of HAV and HEV in the population aged 15–69 years in Peru is high, while the prevalence of HBV and HDV has changed from intermediate to low endemicity level and the prevalence of HCV is low. These findings would prove useful in the development of new strategies aimed at reducing the transmission of viral hepatitis in Peru, with a view to ultimately eliminating these infections in the future.

## Introduction

Viral hepatitis (hepatitis A, B, C, D and E) remains a global public health problem, comparable to other communicable diseases such as human immunodeficiency virus (HIV), tuberculosis and malaria [1]. In 2013, viral hepatitis was ranked the seventh leading cause of mortality worldwide, with an estimated death rate of 1.4 million each year from acute infections and hepatitis-associated liver cancer and cirrhosis [2]. Approximately 47% of deaths are attributable to hepatitis B virus (HBV) and 48% to hepatitis C virus (HCV) infections, and a smaller proportion to hepatitis A (HAV) and E virus (HEV) infections [1]. These infections differ in disease burden and their epidemiology, as well as their clinical and virology features. Due to the high burden of morbidity and mortality, the United Nations, through its health-related Sustainable Development Goals (SDGs), have urged for specific measures to be taken to combat viral hepatitis worldwide [3].

HAV and HEV infections share the fecal–oral route of transmission and generally carry low mortality rates. National surveys in Peru have revealed a high prevalence rate (98%) of HAV infection [4–7], which varies greatly, depending on age and socio-economic level [8]. By contrast, isolated studies of HEV infection have reported high prevalence rates of between 10.4% and 16% [9–11]. However, the national prevalence of HEV infection and how it varies by age, sex, ethnicity and region remains unknown.

HBV and HCV infections can be transmitted parenterally, horizontally or vertically, while HDV is transmitted via simultaneous infection with HBV (that is, co-infection) or as a superimposed infection in a person previously infected with HBV (that is, superinfection). The prevalence rates of HBV and HCV infections vary widely across countries and across regions within a country. The endemicity of HBV infection can be classified into areas of high (prevalence ≥8%), intermediate (prevalence 2–7%) and low endemicity (prevalence <2%). Peru is an intermediate-endemicity country for HBV, with prevalence rates varying widely across its different regions and population groups. For example, in the Peruvian Amazon, the prevalence rates of hepatitis B surface antigen (HBsAg) vary from 2.5% in the Iquitos population to 20% in the indigenous population [12, 13], whereas in the Peruvian coastal regions, the prevalence rates of HBsAg ranges between 1% and 3.5% [12, 14].

In Peru, a pilot vaccination program against hepatitis B was introduced in two hyperendemic provinces (Abancay and Huanta) in 1991 and 1994 for children aged under 5 years, and it was extended in 2003 to include all newborns in HBV-endemic areas. As a result of this intervention, several advances have been made in the control of HBV infection over the last 20 years, including reduced mortality burden from liver diseases associated with HBV infection such as cirrhosis, hepatocellular carcinoma and fulminant hepatitis [15]. According to data from a serological survey in the Peruvian Amazon, the prevalence of HBsAg in children aged under 5 years fell to zero in 2010 [16], and the prevalence of HBsAg in the general population of the highlands decreased from 9.8% to <2% after 20 years of the vaccination program [15].

On the other hand, a serological survey of blood donors in Peru in 2000–2001 revealed prevalence rates of HCV infection ranging between 0.25% and 0.28% [17, 18]. Like HBV infection, HDV infection was reported among native communities in the Peruvian jungle and in some locations in the Peruvian highlands such as Abancay and Huanta where a prevalence of 14% of HDV infection has been reported in apparently healthy school-age children [19]. In addition, studies also found that 17% of individuals with HBV infection and 56.5% of HBsAg carriers had co-infection with HDV [19, 20].

Apart from studies conducted over two decades ago that examined the prevalence rates of HAV, HBV, HCV and HDV infections in the general population and in native populations, including school-age children and blood donors, to the best of our knowledge, to date, there are no updated epidemiological data on viral hepatitis in the different populations and regions of Peru. Data on the prevalence rates of HEV infection are also lacking, since few studies have focused on the epidemiology of HEV infection in Peru. Thus, despite the remarkable progress achieved in HBV prevention in newborns and children, thanks to vaccination, the impact of the vaccination program over the last two decades on endemicity levels of HBV infection in different regions and populations of Peru and on seroprotective levels of hepatitis B surface antibody (anti-HBs) is still unclear. Also uncertain is whether there are still populations susceptible to hepatitis B. Without detailed and precise knowledge of the prevalence of HBV infection in Peru, it is difficult to implement effective strategies aimed at prevention and control with a view to eliminating this infection in the future.

The aim of this study was to determine the prevalence of HAV, HBV, HCV, HDV and HEV infections, as well as the seroprotective levels of anti-HBs, in the Peruvian population. These updated baseline data would aid in the development of strategies aimed at reducing the transmission of viral hepatitis in Peru in the future.

## Materials and methods

### Study design

A cross-sectional, population-based study was conducted between December 2014 and June 2015 in the 25 departments of Peru, a multicultural and multilingual country that is divided into three regions (the coast, the highlands and the jungle). At the time of the study, the total population of Peru, according to the Population and Housing Census of the National Institute of Statistics and Informatics (INEI) in 2007, was 31 million inhabitants.

### Population and sample

The study sample was obtained using multistage probabilistic sampling, according to the sample framework of the Population and Housing Census of the INEI in 2007. The sample size was determined for an infinite population. A 95% confidence level ($z$) was selected, based on a maximum estimation error ($e$) of 5%, a design effect ($deff$) of 1.5 to compensate for the loss of precision as a result of cluster sampling, a prevalence ($p$) of 50% and a response rate of 80% to allow for possible absences and rejections. Thus, we obtained a sample size of 5042 participants, which was distributed proportionally across each region in the country according to the respective regional population size. The work was carried out in collaboration with the offices of statistics and epidemiology of each of the 25 regions of Peru, with demographic information from the INEI giving the total population size of each region in order to determine the sample size required for that region. For each region, a sample size proportional to the region's total population was allocated. In the case of regions with a calculated sample size of <50 participants, a total of 50 participants were included.

### Enrollment

The study included people of both sexes, aged 15–69 years, who had resided for >6 months in a specific region of Peru. One individual from each selected dwelling participated in the study. If there were several families living in a home with individuals aged between 15 and 69 years, one individual from each age group (15–18, 19–29, 30–65 and over 65 years) was included in the study. Demographic, as well as family and community, data and information on the vaccination schedule for hepatitis B were obtained, with prior informed consent from the participants.

### Sample collection

Two 10-mL samples of venous blood (one sample collected in a tube containing EDTA anticoagulant, and the other sample in a tube without EDTA) were obtained from each participant. All samples were sent immediately to each region's local laboratories where they were centrifuged to obtain serum and plasma, which was then stored at –20˚C before being sent first to the Regional Reference Laboratories and then to the National Reference Laboratory of Hepatitis of the National Public Health Center of the National Institute of Health of Peru. Samples were transported in cold chain at temperatures ranging between 4˚C and 8˚C, in accordance with the standards of biosecurity and transport of biological samples of category "B".

Serum samples were analyzed for the following serological markers: immunoglobulin G (IgG) antibodies against HAV (anti-HAV IgG), HBsAg, immunoglobulin M (IgM) and IgG antibodies against HBV core antigen (anti-HBc IgM and IgG), anti-HBs quantitative, antibodies against HCV (total anti-HCV), antibodies against HDV and antibodies against HEV (anti-HEV IgG). HBsAg and total anti-HBc IgG were analyzed using the chemiluminescence method with the Maglumi reagent (Snibe Diagnostic, Guangdong, China). The sensitivity of HBsAg is 99.88% and the specificity is 99.75%, the sensitivity of anti-HBc IgG is 99.92% and the specificity is 99.82%. Anti-HAV IgG, anti-HCV, anti-HEV IgG and anti-HBs were analyzed by enzyme-linked immunosorbent assay (ELISA) (Beijing Wantai Biological Pharmacy, Beijing, China). Of note, the sensitivity of anti-HAV IgG is 100% and the specificity is 99%, the sensitivity of anti-HCV is 99.88% and the specificity is 99.87%, the sensitivity of anti-HEV IgG is 99% and the specificity is 99.99%, and the sensitivity of anti-HBs is 100% and the specificity is 99.58%. The reactive HBsAg samples were analyzed by ELISA (Beijing Wantai Biological Pharmacy) for HBeAg, anti-HBc IgM (99.84% sensitivity and 99.93% specificity), antibodies against HBV envelope antigen (anti-HBe), anti-HDV IgM and anti-HDV IgG (100% sensitivity and 100% specificity). In chronic HBV carriers, the viral load (DNA levels) was determined by real-time quantitative polymerase chain reaction (qPCR) using the COBAS AmpliPrep/ COBAS TaqMan VHB Test kits, version 2.0 (Roche Molecular Diagnostics, Branchburg, NJ, USA), which has a lower limit of detection of 20 IU/mL.

Chronic HBV infection was defined as the presence of HBsAg and total anti-HBc (IgG) and the absence of anti-HBc IgM. Past infection was defined as the presence of total anti-HBc IgG and the absence of HBsAg and anti-HBc IgM. Acute infection was defined as the presence of HBsAg, total anti-HBc and anti-HBc IgM. Levels of anti-HBs (quantitative) of ≥10 mUI/ml and <10 mUI/ml were considered as protective and non-protective against HBV infection, respectively.

### Statistical analysis

Analyses were performed using SPSS Software version 21.0 (SPSS, Inc., Chicago, IL, USA). Prevalence rates (by sex, age, region, department, ethnicity and occupation) of anti-HAV, HBsAg, anti-HBc IgG and IgM, anti-HBs, anti-HCV, anti-HDV and anti-HEV were

determined, with 95% confidence intervals (CIs). The prevalence rates were compared using Pearson's chi-squared test or Fisher's exact test.

## Ethics

All participants aged over 18 years provided written informed consent before their enrollment in the study. For participants aged under 18 years, informed consent was obtained from their parents or guardians. This study was approved by the Ethics and Research Committee of the National Institute of Health of Peru.

## Results

A total of 5183 participants aged 15–69 years from 25 regions of Peru were included in the study. Of these, 71.5% were women and 54.3% were aged between 30 and 65 years (Table 1).

### Prevalence of hepatitis A

Of the total of 5183 participants, 5099 tested positive for anti-HAV IgG (98.4%, 95% CI 98.0–98.7). The prevalence rate of anti-HAV IgG was similar in both men and women (P = 0.813), and significantly higher among participants aged between 30 and 65 years (99.4%; P<0.0001). The prevalence rate was also similar across all ethnic groups (P = 0.537) and across different educational levels (P = 0.377) (Table 1).

The prevalence rates of anti-HAV IgG in the coastal, highland and jungle regions, as well as across the 25 departments of Peru, were high (≥90%), except in the Pasco Department where the prevalence rate was lower (82.8%, 95% CI 72.8–92.8) (Table 2). Within each age group, there were similar regional differences in the prevalence rate of anti-HAV IgG, although the prevalence increased with increasing age (Fig 1A). The same overall trend was observed when the prevalence rates of anti-HAV IgG in urban and rural areas were compared within each age group and with increasing age (Fig 2A).

### Prevalence of hepatitis B and D

Of the 5183 participants, 20 were positive for HBsAg (0.4%, 95% CI 0.21–0.55), 526 for total anti-HBc IgG (10.1%, 95% CI 9.3–10.9) and 3104 for anti-HBs ≥10 mUI/ml (60%, 95% CI 58.5–61.2). All HBsAg-positive participants were chronic inactive carriers of HBV (anti-HBe positive, and HBeAg and anti-HBc IgM negatives). Viral loads were <2000 mIU/mL in all participants. Viral loads of between 1000–1999 mIU/mL were found in two carriers, 20–999 mIU/mL in 11 carriers and <20 mIU/mL in four carriers, while three had undetectable levels. The prevalence of HDV among HBsAg carriers was 15% (3/20).

The prevalence of HBsAg and anti-HBc IgG was significantly higher in men than in women (HBsAg, 0.7% vs 0.3%, P<0.033; anti-HBc IgG, 13% vs 9%, P<0.0001), whereas the prevalence of anti-HBs ≥10 mUI/ml was similar in both sexes (P<0.293). The prevalence of anti-HBc IgG increased significantly with age (P<0.0001), whereas there was a significant decrease in the prevalence of anti-HBs ≥10 mUI/ml with increasing age (P<0.0001). The prevalence rates of HBsAg across the four ethnic groups studied were similar, but the prevalence rates of anti-HBc IgG and anti-HBs ≥10 mUI/ml were significantly higher among indigenous/Ashianinka (25.7%; P<0.0001) and Afro-Peruvian (82.4%; P<0.0001) groups, respectively. In addition, the prevalence rates of anti-HBc IgG and anti-HBs ≥10 mUI/ml were also significantly higher among health workers (76.4%; P<0.0001) (Table 1).

The prevalence rates of HBsAg, anti-HBc IgG and anti-HBs ≥10 mUI/ml by geographic distribution are shown in Table 2. On average, the prevalence rates of HBsAg in the coastal,

**Table 1. Sociodemographic characteristics and prevalence of hepatitis A, B, C and E.**

| | Total | Anti-HAV | $p^*$ | HBsAg | $p^*$ | Anti-HBc | $p^*$ | Anti-HCV | $p^*$ | Anti-HBs ≥10 mUI/ml | $p^*$ | Anti-HEV | $p^*$ |
|---|---|---|---|---|---|---|---|---|---|---|---|---|---|
| | n (%) | n (%) | | n (%) | | n (%) | | n (%) | | n (%) | | n (%) | |
| **Sex** | | | | | | | | | | | | | |
| Male | 1479 (28.5) | 1456 (98.4) | 0.813 | 10 (0.7) | 0.033 | 193 (13) | 0.0001 | 0 (0.0) | 0.121 | 869 (59) | 0.293 | 204 (13.8) | 0.722 |
| Female | 3704 (71.5) | 3643 (98.4) | | 10 (0.3) | | 333 (9.0) | | 6 (0.2) | | 2235 (60.3) | | 525 (14.0) | |
| **Age**[**] | | | | | | | | | | | | | |
| **15–18** | 531 (10.2) | 497 (93.6) | <0.0001 | 2 (0.4) | 0.763 | 19 (3.6) | <0.0001 | 0 (0.0) | 0.731 | 458 (86.3) | <0.0001 | 35 (6.6) | <0.0001 |
| **19–29** | 1716 (33.1) | 1684 (98.1) | | 5 (0.3) | | 124 (7.2) | | 3 (0.2) | | 1253 (73.0) | | 142 (8.3) | |
| **30–65** | 2816 (54.3) | 2799 (99.4) | | 12 (0.4) | | 359 (12.7) | | 3 (0.1) | | 1354 (48.1) | | 509 (18.1) | |
| **>65** | 119 (2.3) | 118 (99.2) | | 1 (0.8) | | 24 (20.2) | | 0 (0.0) | | 39 (32.8) | | 43 (36.1) | |
| **Ethnicity**[***] | | | | | | | | | | | | | |
| Quechua | 325 (6.8) | 321 (98.8) | 0.537 | 4 (1.2) | 0.237 | 62 (19.1) | <0.0001 | 0 (0.0) | 0.995 | 199 (61.2) | <0.0001 | 42 (12.9) | 0.519 |
| Aymara | 82 (1.7) | 80 (97.6) | | 1 (1.2) | | 8 (9.8) | | 0 (0.0) | | 34 (41.5) | | 6 (7.3) | |
| Indigenous/Ashaninka | 109 (2.3) | 108 (99.1) | | 1 (0.9) | | 28 (25.7) | | 0 (0.0) | | 75 (68.8) | | 11 (10.1) | |
| Mixed | 4139 (86.1) | 4070 (98.3) | | 13 (0.3) | | 385 (9.3) | | 6 (0.1) | | 2499 (60.4) | | 558 (13.5) | |
| Afro-Peruvian | 17 (0.4) | 17 (100) | | 0 (0.0) | | 3 (17.6) | | 0 (0.0) | | 14 (82.4) | | 4 (23.5) | |
| European/White | 30 (0.6) | 28 (93.3) | | 0 (0.0) | | 2 (6.7) | | 0 (0.0) | | 23 (76.7) | | 3 (10) | |
| Oriental | 3 (0.1) | 3 (100) | | 0 (0.0) | | 0 (0.0) | | 0 (0.0) | | 2 (66.7) | | 0 (0.0) | |
| Other | 102 (2.1) | 100 (98.0) | | 0 (0.0) | | 11 (10.8) | | 0 (0.0) | | 72 (70.6) | | 12 (11.8) | |
| **Education level**[¶] | | | | | | | | | | | | | |
| Incomplete Primary | 307 (5.9) | 304 (99) | 0.377 | 3 (1.0) | 0.155 | 65 (21.2) | <0.0001 | 0 (0.0) | 0.931 | 155 (50.5) | <0.0001 | 69 (22.5) | <0.0001 |
| Primary | 545 (10.6) | 540 (99.1) | | 2 (0.4) | | 107 (19.6) | | 1 (0.2) | | 265 (48.6) | | 89 (16.3) | |
| Incomplete secondary | 575 (11.1) | 565 (98.3) | | 3 (0.5) | | 77 (13.4) | | 1 (0.2) | | 340 (59.1) | | 89 (15.5) | |
| Secondary | 1564 (30.3) | 1542 (98.6) | | 2 (0.1) | | 131 (8.4) | | 2 (0.1) | | 880 (56.3) | | 230 (14.7) | |
| Incomplete higher education | 499 (9.7) | 488 (97.8) | | 4 (0.8) | | 33 (6.6) | | 0 (0.0) | | 308 (61.7) | | 53 (10.6) | |
| Higher education | 1673 (32.4) | 1640 (98) | | 6 (0.4) | | 107 (6.4) | | 2 (0.1) | | 1147 (68.6) | | 192 (11.5) | |
| **Occupation** [¶¶] | | | | | | | | | | | | | |
| Health worker | 526 (10.3) | 519 (98.7) | <0.0001 | 3 (0.6) | 0.089 | 50 (9.5) | <0.001 | 0 (0.0) | 0.332 | 402 (76.4) | <0.0001 | 62 (11.8) | <0.007 |
| Housewife | 1072 (21.0) | 1064 (99.3) | | 2 (0.2) | | 130 (12.1) | | 3 (0.3) | | 529 (49.3) | | 174 (16.2) | |
| Other activity | 2628 (51.4) | 2593 (98.7) | | 7 (0.3) | | 283 (10.8) | | 2 (0.1) | | 1506 (57.3) | | 381 (14.5) | |
| Not working | 886 (17.3) | 854 (96.4) | | 7 (0.8) | | 59 (6.7) | | 1 (0.1) | | 620 (70) | | 101 (11.4) | |

[*]χ2 test

[**]Data on age were available for 5182 of 5183 participants included in the study

[***] Data on Ethnicity were available for 4807 of 5183 participants included in the study

[¶] Data on Education level were available for 5163 of 5183 participants included in the study

[¶¶] Data on occupation were available for 5112 of 5183 participants included in the study

**Table 2. Prevalence of hepatitis A, B, C and E in the different regions of Peru, 2014–2015.**

| | Total | Anti-HAV | | HBsAg | | Anti-HBc | | Anti-HBs (≥10 mUI/ml) | | Anti-HCV | | Anti-HEV IgM | |
|---|---|---|---|---|---|---|---|---|---|---|---|---|---|
| | n (%) | N | % (CI 95%) | N | % (CI 95%) | n | % (CI 95%) | n | % (CI 95%) | N | % (CI 95%) | n | % (CI 95%) |
| **Coast** | 2883 | 2853 | 99.0% (98.5–99.5) | 7 | 0.2% (0.06–0.40) | 149 | 5.2% (4.4–6.0) | 1683 | 58.4% (56.6–60.2) | 3 | 0.1% (-0.01–0.2) | 496 | 17.6% (16.2–19) |
| Lima | 1616 | 1608 | 99.0% (98.0–99.4) | 4 | 0.2% (0.04–0.49) | 100 | 6.2% (5.0–7.4) | 1014 | 62.7% (60.4–65.1) | 2 | 0.1% (-0.04–0.3) | 283 | 18.3% (16.3–20.2) |
| La Libertad | 300 | 297 | 99% (97.8–100) | 0 | 0% (0.0) | 7 | 2.3% (0.6–4.0) | 162 | 54% (48.3–59.7) | 0 | 0% (0.0) | 44 | 14.6% (10.6–18.7) |
| Piura | 296 | 292 | 98.6% (97.0–99.8) | 0 | 0% (0.0) | 7 | 2.4% (0.6–4.1) | 186 | 62.8% (57.3–68.4) | 0 | 0% (0.0) | 53 | 17.9% (13.5–22.3) |
| Ica | 133 | 132 | 99.2% (97.8–100) | 1 | 0.75% (-0.7–2.2) | 5 | 3.8% (0.5–7.0) | 54 | 40.6% (32.1–49.0) | 0 | 0% (0.0) | 20 | 15.0% (8.8–21.2) |
| Lambayeque | 219 | 217 | 99.0% (97.8–100) | 0 | 0% (0.0) | 11 | 5.0% (2.1–7.9) | 122 | 55.7% (49.0–62.3) | 0 | 0% (0.0) | 37 | 16.9% (11.9–21.9) |
| Callao | 161 | 157 | 98.8% (96.3–100) | 2 | 1.24% (-0.4–2.97) | 8 | 4.96% (1.6–8.4) | 71 | 44.0% (36.3–51.9) | 0 | 0% (0.0) | 36 | 22.4% (15.9–28.9) |
| Tumbes | 47 | 47 | 100% | 0 | 0% (0.0) | 8 | 17.0% (5.8–28.2) | 21 | 44.7% (30–59) | 0 | 0% (0.0) | 4 | 8.5% (0.2.17) |
| Tacna | 56 | 52 | 100% (99–100) | 0 | 0% (0.0) | 2 | 3.6% (-1.4–8.5) | 24 | 42.9% (29–56) | 1 | 1.8% (-1.7–5.4) | 7 | 12.5% (3.6–21.4) |
| Moquegua | 55 | 52 | 100% (99–100) | 0 | 0% (0.0) | 1 | 1.8% (-1.8–5.4) | 29 | 53.0% (39–66) | 0 | 0% (0.0) | 12 | 21.8% (10.6–33) |
| **Highlands** | 1790 | 1765 | 98.9% (98.3–99.5) | 10 | 0.7% (0.3–1.0) | 281 | 15.8% (14.1–17.5) | 1100 | 61.5% (59.1–63.7) | 1 | 0.02% (-0.07–0.4) | 206 | 11.6% (10.1–13.1) |
| Cusco | 223 | 222 | 99.6% (98.7–100) | 5 | 3.1% (0.51–5.8) | 101 | 45.3% 8 (38.7–51.9) | 193 | 86.5% (82.0–91.0) | 0 | 0% (0.0) | 19 | 8.5% (4.8–12.2) |
| Ayacucho | 112 | 110 | 98.2% (95.7–100) | 2 | 1.78% (-0.7–4.2) | 58 | 51.7% (42.3–61.2) | 85 | 75.9% (67.8–83.9) | 0 | 0% (0.0) | | 5.3% (1.1–9.6) |
| Pasco | 58 | 48 | 82.8% (72.8–92.8) | 0 | 0% (0.0) | 20 | 34.5% (21.9–47.0) | 52 | 89.7% (81.6–97.7) | 0 | 0% (0.0) | 6 | 10.3% (2.3–18.4) |
| Huánuco | 122 | 119 | 97.5% (94.8–100) | 0 | 0% (0.0) | 31 | 25.4% (17.6–33.2) | 88 | 72.1% (64.0–80.2) | 0 | 0% (0.0) | 5 | 9.0% (3.9–14.2) |
| Junin | 216 | 214 | 99.0% (97.8–100) | 1 | 0.5% (-0.4–1.4) | 24 | 11.1% (6.9–15.3) | 133 | 61.6% 8 (55.0–68.1) | 0 | 0% (0.0) | 16 | 7.4% (3.9–10.9) |
| Cajamarca | 256 | 255 | 99.6% (98.8–100) | 0 | 0% (0.0) | 17 | 6.6% (3.5–9.7) | 129 | 50.3% (44.2–56.5) | 0 | 0% (0.0) | 17 | 6.6% (3.5–9.7) |
| Arequipa | 213 | 212 | 99% (99.5–100) | 0 | 0% (0.0) | 8 | 3.7% (1.18–6.3) | 86 | 40.3% (33.7–47.0) | 0 | 0% (0.0) | 63 | 29.5% (23.4–35.7) |
| Ancash | 195 | 195 | 100% | 0 | 0% (0.0) | 1 | 0.51% (0.4–1.5) | 102 | 52% (45–59) | 1 | 0.51% (0.4–1.5) | 31 | 15.8% (11–21) |
| Apurímac | 81 | 100 | 100% | 1 | 1.23% (-1.2–3.6) | 8 | 9.8% (16.2–3.5) | 60 | 74% (64–84) | 0 | 0% (0.0) | 8 | 9.8% (3.2–16.5) |
| Huancavelica | 85 | 84 | 98.8% (96.5–100) | 0 | 0% (0.0) | 5 | 8.23% (1.4–15.0) | 60 | 70.6% (60.7–80.5) | 0 | 0% (0.0) | 10 | 14.1% (5.9–22.4) |
| Puno | 229 | 228 | 99.6% (98.7–100) | 1 | 0.43% (-0.4–1.3) | 8 | 3.5% (1.0–5.9) | 112 | 49.0% (42.4–55.4) | 0 | 0% (0.0) | 19 | 8.3% (4.7–11.9) |
| **Jungle** | 510 | 501 | 99.8% (98.6–100) | 3 | 0.6% (-0.07–1.2) | 96 | 19.2% (15.7–22.7) | 321 | 63.0% (58.7–67.1) | 2 | 0.04% (-0.02–0.9) | 27 | 5.3% (3.3–7.2) |
| San Martín | 130 | 128 | 98.5% (96.3–100) | 0 | 0% (0.0) | 12 | 9.2% (4.2–14.3) | 91 | 70% (62.0–78.0) | 1 | 0.8% (-0.8–2.3) | 9 | 7.0% (2.5–11.3) |
| Ucayali | 78 | 78 | 100% | 0 | 0% (0.0) | 29 | 37.2% (26.2–48.1) | 67 | 86.0% (78.0–93.8) | 0 | 0% (0.0) | 2 | 2.6% (-1.0–6.1) |
| Loreto | 160 | 152 | 100% | 2 | 1.25% (-0.4–2.9) | 24 | 16.3% (10.2–22.3) | 67 | 42.0% (34.1–49.6) | 1 | 0.63% (-0.6–1.8) | 10 | 6.3% (2.5–10.0) |
| Madre de Dios | 60 | 60 | 100% | 0 | 0% (0.0) | 9 | 15% (5.6–24.3) | 38 | 63.3% (50.8–75.9) | 0 | 0% (0.0) | 3 | 5% (-0.6–10.7) |
| Amazonas | 82 | 81 | 98.8% (96.4–100) | 1 | 1.2% (-1.2–3.6) | 22 | 26.8% (17.0–36.6) | 58 | 70.7% (60.7–80.8) | 0 | 0% (0.0) | 3 | 3.7% (-0.4–7.8) |

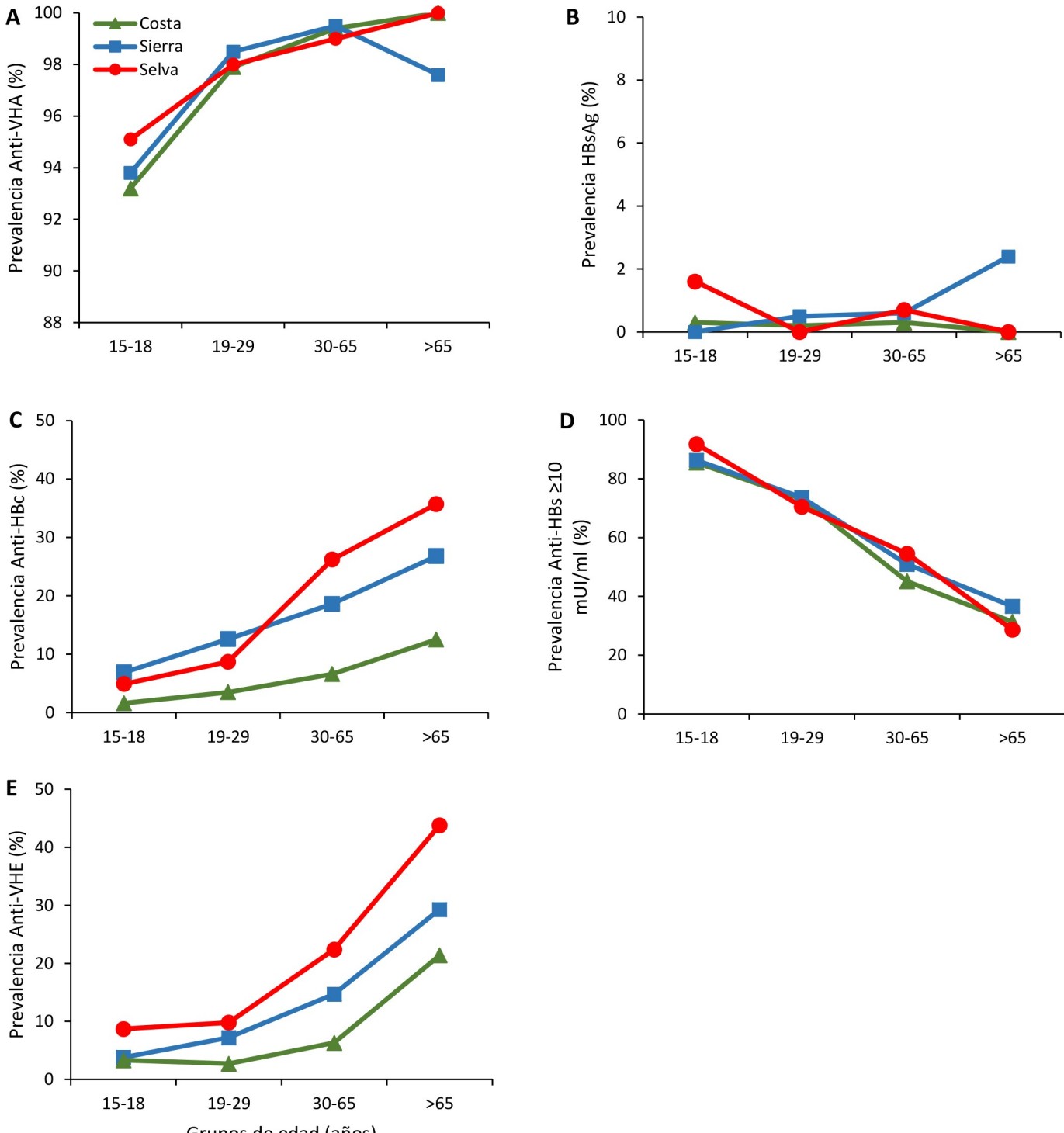

**Fig 1.** Prevalence rates of anti-HAV IgG (A), HBsAg (B), anti-HBc IgG (C), anti-HBs ≥10 mUI/ml (D) and anti-HEV IgG (E) by age groups in different regions of Peru.

highland and jungle regions, as well as in the 25 departments, of Peru were low, except for the Cusco region where the prevalence of HBsAg was intermediate (3.1%, 95% CI 0.51–5.8). Fig 3A

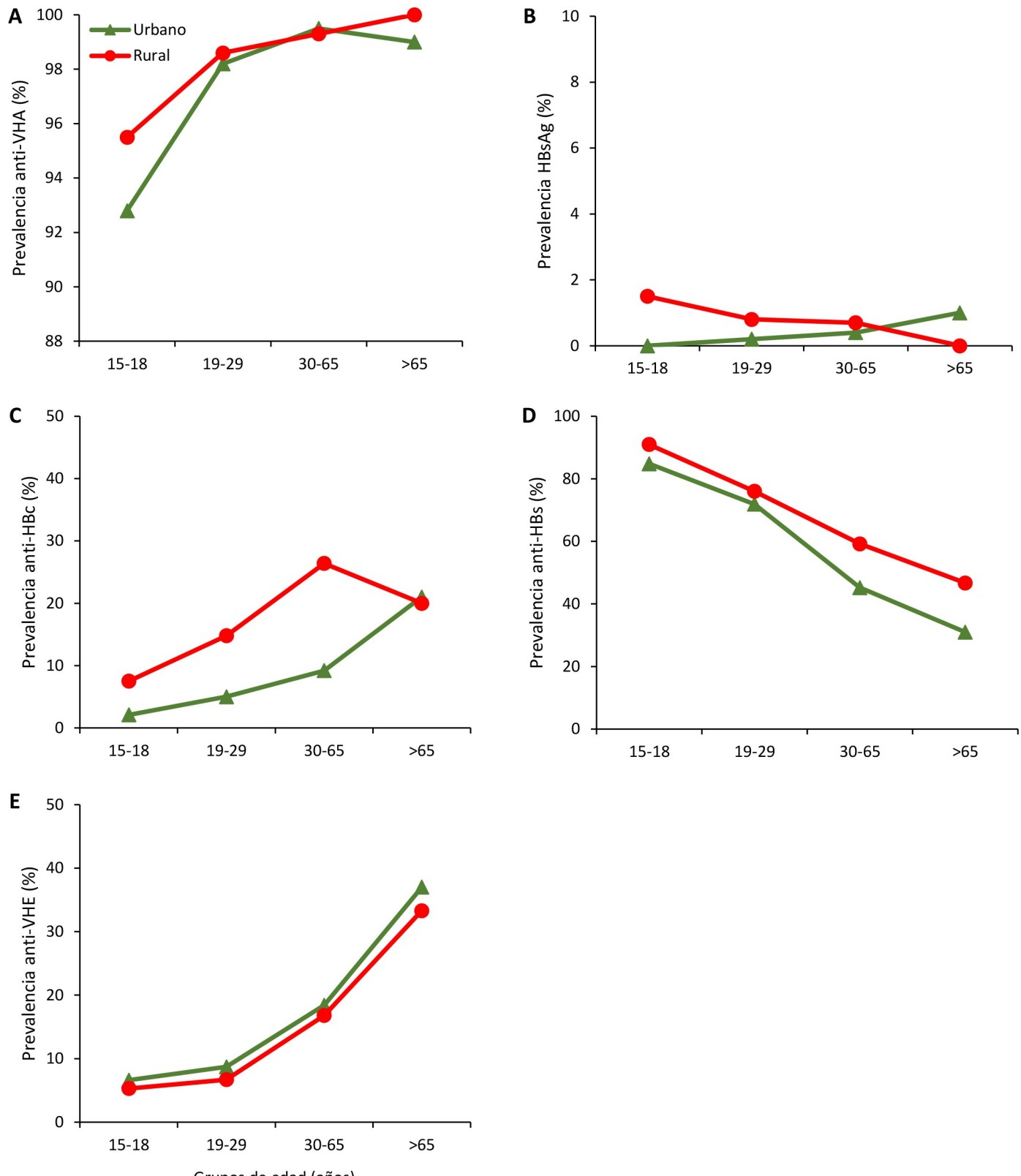

**Fig 2.** Prevalence rates of anti-HAV IgG (A), HBsAg (B), anti-HBc IgG (C), anti-HBs ≥10 mUI/ml (D) and anti-HEV IgG (E) by age groups in urban and rural areas of Peru.

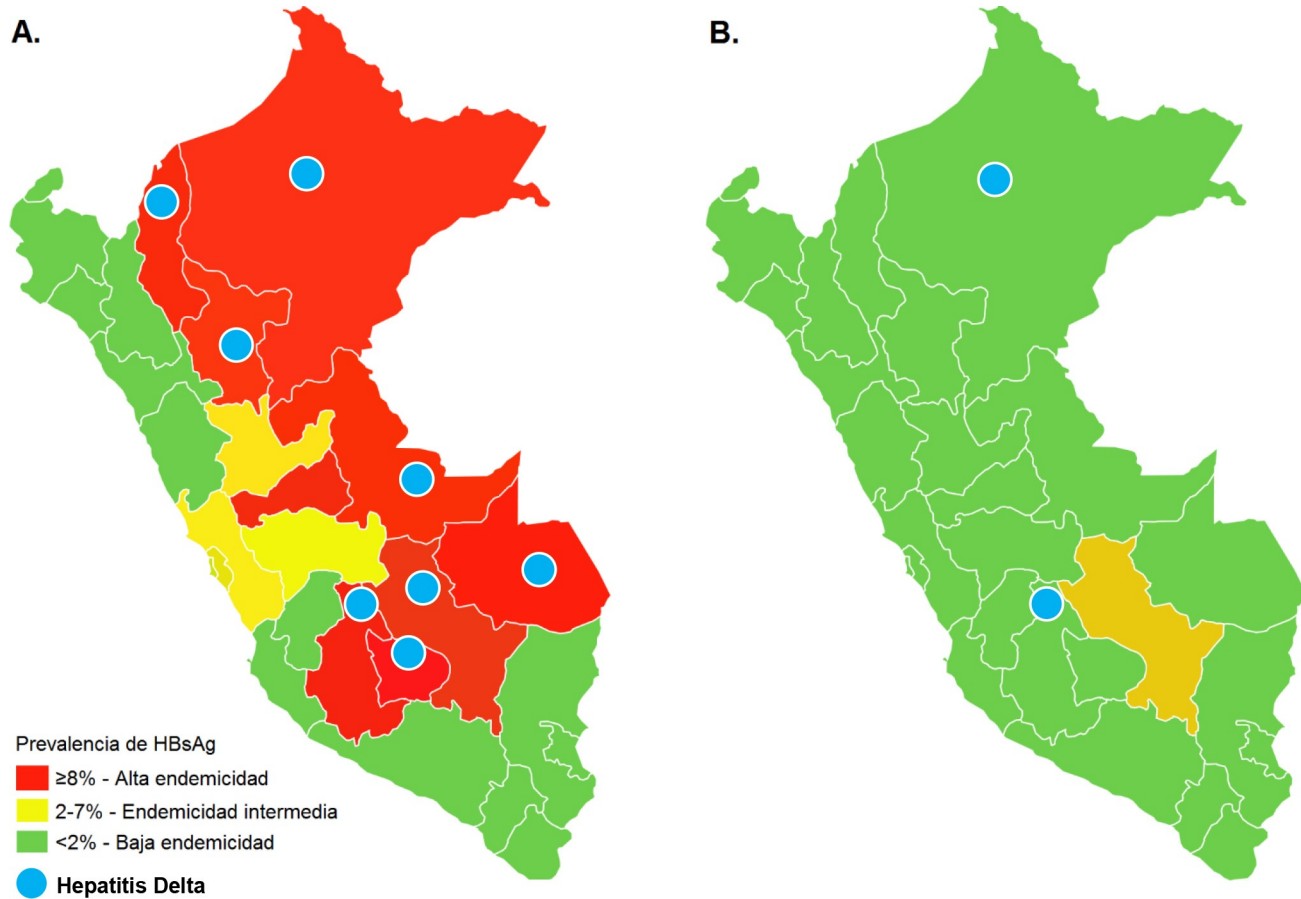

**Fig 3.** Prevalence of HBsAg and hepatitis Delta before (A) [19] and after the implementation of the hepatitis B vaccination program in Peru (B).

and 3B show the change in HBsAg and HDV endemicity pattern before and after the implementation of the hepatitis B vaccination program in Peru. Of note, HDV now circulates in only two departments of Peru, namely Ayacucho and Loreto (Fig 3B), compared to the pre-vaccination era (Fig 3A). Fig 4 show the prevalence rates of anti-HBc IgG after the implementation of the hepatitis B vaccination program in Peru.

With respect to age groups, regional differences in HBsAg prevalence rates were more evident among participants aged 15–18 years (Figs 1B and 2B). There were regional differences in the prevalence rates of anti-HBc IgG for all age groups, with an increase in prevalence with increasing age (Figs 1C and 2C). By contrast, the prevalence rates of seroprotective anti-HBs (≥10 mIU/mL) were similar across regions and decreased with increasing age (Figs 1D and 2D).

## Prevalence of hepatitis C

Only six of 5183 participants tested positive for anti-HCV, thus indicating a low prevalence of anti-HCV (0.1%, 95% CI 0.02–0.2). Positive anti-HCV were only found in women (0.2%) aged 19–29 years (0.2%) and 30–65 years (0.1%) of mixed race (0.1%), with similar rates obtained across different educational levels and occupational activities (Table 1). HCV infection was only found in five departments of Peru: Lima, Tacna, Ancash, San Martín and Loreto (Table 2).

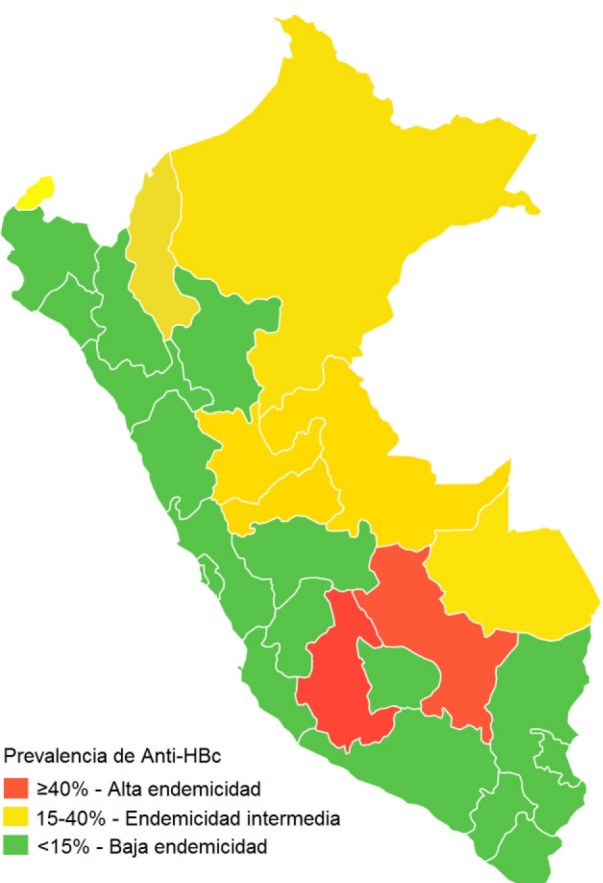

Prevalencia de Anti-HBc

- 🟥 ≥40% - Alta endemicidad
- 🟨 15-40% - Endemicidad intermedia
- 🟩 <15% - Baja endemicidad

**Fig 4. Prevalence of anti-HBc IgG after the implementation of the hepatitis B vaccination program in Peru.**

## Prevalence of hepatitis E

A total of 729 of 5183 participants tested positive for anti-HEV IgG (14%, 95% CI 13.1–15.0). The prevalence of anti-HEV IgG was similar between men and women (P = 0.722) and increased significantly with age (P<0.0001). There was a higher prevalence among Afro-Peruvians (23.5%), and significantly higher prevalence rates among people who had not completed primary education (22.5%; P<0.0001) as well as among housewives (16.2%; P = 0.007) (Table 1).

Anti-HEV IgG prevalence was higher in the coastal region (14%, 95% CI 13.1–15.0), compared to the highlands and the jungle. Moreover, in the coastal, highland and jungle regions, there were higher prevalence rates in El Callao (22.4%, 95% CI 15.9–28.9), Arequipa (29.5%, 95% CI 23.4–35.7) and San Martín (7.0%, 95% CI 2.5–11.3) (Table 2 and Fig 5). Across all age groups, regional differences in the prevalence of anti-HEV were evident and these prevalence rates increased with age (Figs 1E and 2E).

All HBsAg-positive and anti-HCV-positive participants were also anti-HAV-positive. The prevalence of anti-HAV IgG among anti-HEV-positive participants was high (99.3%, 724/729), whereas there was a low prevalence of HBsAg among anti-HEV-positive participants (0.55%, 4/729). One participant was HBsAg-, VHD- and anti-HEV-positive, and another was anti-HAV-, anti-HCV- and anti-HEV-positive.

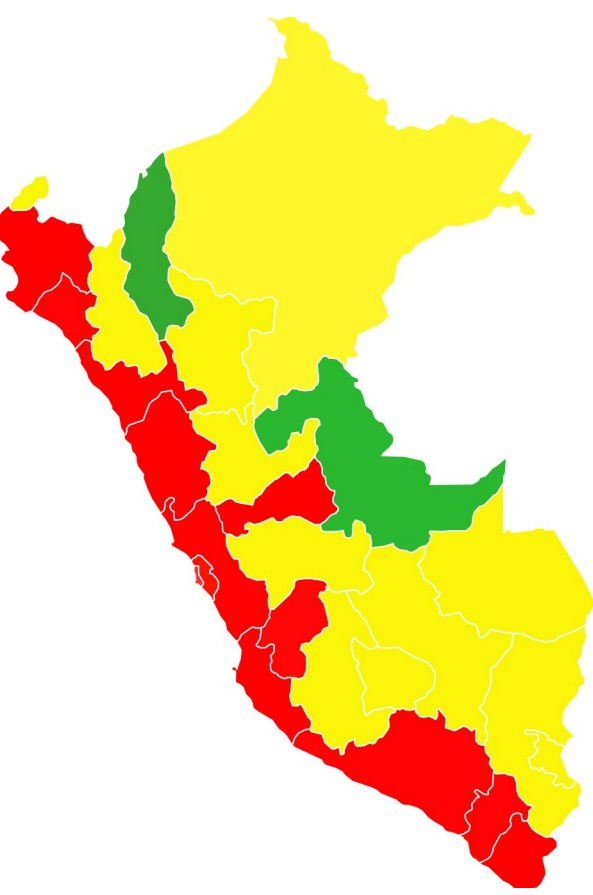

**Fig 5. Prevalence of anti-HEV IgG in Peru, 2014–2015.**

## Discussion

This study presented results from the first national survey of the prevalence of HAV, HBV, HCV, HDV and HEV infections, as well as the seroprotective levels of anti-HBs, in Peru. Our findings showed a high prevalence of anti-HAV (98.38%), consistent with previous studies on the prevalence of anti-HAV in coastal regions (Lima, 91% of adults) and the Peruvian jungle (Loreto, 98% of adults) [8]. Other groups also reported high prevalence rates of anti-HAV in children in the Peruvian highlands, notably in Huanta (98%) [5] and Huanuco (95.2%) [6], as well as in other countries, including Brazil (95% in Chaco and 86.4% in Mata Grosso) [21, 22]. In contrast to our findings, a study conducted in the pediatric population of five regions of Peru [7] and another study in subjects aged 1–39 years presenting with symptoms compatible with HAV infection [8] showed lower prevalence rates of 50.5% and 65.2%, respectively. In Santiago de Chile, a prevalence rate of anti-HAV of 40.6% was also found in the population aged 1–24 years [23]. These variations in prevalence rates of anti-HAV IgG across different studies are probably related to differing hygiene habits, age groups studied, educational levels and local water and sanitation conditions.

In our study, we must highlight that the regions of Apurímac, Madre de Dios, Tumbes and Ucayali had the highest prevalence rates of anti-HAV IgG, compared to the region of Pasco (82.8%). This regional difference probably reflects the different extent and ways of improvement in hygiene habits and health conditions across the various regions of Peru. Moreover, we found a higher prevalence of anti-HAV IgG in Lima (98.2%), compared to that previously

reported by Kilpatrick (92%) [4] and Hernandez (37.4%) [7]. Such discrepancy can be explained by the constant migration of populations, particularly in the age groups studied, from the interior of the country to Lima, hygiene habits, educational levels. We must also consider that, unlike in the metropolitan area of Lima, there are locations in the peri-urban areas of Lima that lack adequate water and sanitation facilities.

Previously, Peru was a country with an intermediate endemicity level for HBV infection, presenting with areas of high, medium and low endemicity in its different regions. Added to this has been the effect of internal migration [14, 19]. In this study, we found a change in the level of endemicity for HBV infection from intermediate to low, as evidenced by the low prevalence of HBsAg (0.38%) and total anti-HBc (10.14%). Our findings also showed a reduction in HBsAg rates, indicating a low endemicity level, in the previously hyperendemic areas of the Peruvian Amazon and the inter-Andean valleys of the highlands (Ayacucho and Apurímac). These findings are compatible with the impact of the vaccination program against HBV that the MINSA (Ministry of Health, Peru) launched in 1991–1994 in the inter-Andean valleys of the Peruvian highlands (Abancay and Huanta), followed in 1996 by the implementation of vaccination for children aged under 1 year living in intermediate- and high-endemic areas and then in 2003 with universal vaccination for all children aged under 1 year in Peru [15, 24]. Changes in endemicity levels from high or intermediate to low as a result of universal vaccination against HBV have also been reported in countries, including China, Italy, Australia, Gambia and Colombia [25]. It is also important to highlight that only 60% of the population in our study had seroprotective levels of anti-HBs ≥10 mUI/ml. Given that seroprotection decreases with age, this means there are a significant number of people in Peru who are susceptible to HBV infection and for whom vaccination should be considered, while for those who were previously immunized, revaccination should be considered, as recommended by others in a previous study in China [26].

Significant rates of HBV and HDV co-infection were previously reported in the Amazon and inter-Andean valleys of Peru (Huanta, Abancay) [5, 13]. We found a low prevalence of anti-HDV among HBsAg carriers. These results showed a decrease in the prevalence of HDV infection, in contrast to previous reports from Peru [15, 16, 24] and other countries, including Italy [27], although co-infection is still highly prevalent in some parts of Africa [28] and Colombia [29]. This reduction in the prevalence of HDV infection is also the result of universal vaccination against HBV in Peru, since reducing the rate of HBsAg carriers will decrease the rates of HBV and HDV co-infection.

In the present study, we found a low anti-HCV prevalence (0.12%) in the general population, with rates of <1% in the coastal region, the highlands and the Peruvian jungle. Studies on the prevalence of HCV infection in Peru are scarce and these showed low prevalence rates, except in high-risk groups [17, 18, 30, 31]. A national serological survey of blood donors found anti-HCV prevalence rates of 0.25% in 2000 and 0.60% in 2001, with the highest rates in the coastal region and the Peruvian jungle [17]. Another study of >15,000 blood donors from a hospital in Lima found an anti-HCV prevalence rate of 0.8% [18]. The few studies conducted in at-risk populations found HCV prevalence varying within and across the different groups: hemodialysis patients (15.4–21.6%), hemophiliacs (1.7–7.9%), patients with acute (from 3.6–6.9%) or chronic hepatitis (3.6–0.8%) and intravenous drug users (14.1%) [32]. Highly varied prevalence rates of anti-HCV have also been reported in other Latin American countries. For instance, in Colombia, the prevalence of anti-HCV was found to be 5.68% in the Amazon and 0.66% in San Andres Islands [33]; in Mexico, the national prevalence was reported to be 1.4% [34], while in Buenos Aires (Argentina), the anti-HCV prevalence was found to be 54.6% in intravenous drug users and 7.5% in homosexual men [35]. Similar to the situation in Peru,

these studies showed that there is a low prevalence of HCV infection in the general population, but a high prevalence in high-risk groups such as intravenous drug users.

There are only a limited number of studies on the prevalence of anti-HEV IgG in Peru [9, 10, 11]. In our study, we investigated, for the first time, the prevalence of anti-HEV in the general population of Peru. Our results showed a high anti-HEV prevalence (14%) in the general population that increased with age. These findings are similar to a previous study conducted in two regions of Peru (Lima and Loreto) that reported an 15% of anti-HEV IgG positive, with a rate of 14% among apparently healthy soldiers who had been living for 1–2 years in the Amazon rainforest [10]. Another study in men aged 25–66 years who worked in the drinking water and sewerage system in Lima also reported a high anti-HEV prevalence of 10.4% [9]. Unlike our findings, low prevalence rates of anti-HEV have been reported in other Latin American countries. For instance, in Medellin (Colombia), a prevalence rate of 11.2% was found in 20- to 62-year-old workers of a pig farm [36], while in Venezuela, there was a prevalence of 1.6% in pregnant women, 3.9% in rural residents and 5.4% in rural Amerindians, and a global prevalence of 3.8% [37]. In some health facilities in Brazil, a high prevalence of anti-HEV (10.2%) was found, which was associated with contact with pigs and an increasing prevalence with advanced age [38]. In another study carried out in Rio de Janeiro, a low anti-HEV prevalence was found among patients with acute hepatitis (2.1%), those from rural areas (2.1%) and hemodialysis patients (6.2%), and a high anti-HEV prevalence was found in intravenous drug users (11.8%) [39].

This study has some limitations. First, the present study only included patients aged over 15 years; this was to ensure voluntary participation. The infant population was not considered, since cases of HBV, HCV, HDV and HEV infections have rarely been reported in the pediatric population in recent years, in contrast to the situation with HAV infection. In addition, in order to assess the impact of hepatitis B vaccination at a national level, the pediatric population was not included due to the high vaccination coverage that has been achieved as a result of the implementation of the universal vaccination program in children since 2003. Second, the study population consisted predominantly of women, which was probably due to the fact that in urban and rural areas, women stay longer at home than men. However, our findings should not be affected by this, since prevalence to hepatitis HAV, HBV and HEV infections was similar in both sexes. It is also important to highlight that we used the Wantai commercial kit to determine the prevalence of anti-HEV. One study reported that use of assay kits from different manufacturers can result in underestimation of the actual anti-HEV prevalence [40], whereby it was found that 16.2% of patients were positive for anti-HEV using the Wantai commercial kit, compared to 3.6% of positive patients using the Genelabs commercial kit. These findings indicated the high sensitivity and specificity of the Wantai commercial kit [40].

In conclusion, there is a high prevalence of HAV and HEV among those aged 15–69 years in Peru (hyperendemic), whereas the prevalence of HBV and HDV has changed from intermediate to low as a result of the universal vaccination program against HBV. In addition, the prevalence of HCV remains low in the general population studied. These findings are of significance, particularly in the development of new strategies aimed at reducing the transmission of viral hepatitis in Peru, with a view to ultimately eliminating these infections in the future. Therefore, we recommend that the wide immunization coverage against HBV should be maintained, along with improvement of hygiene, water and sanitation conditions to reduce the high prevalence rates of HAV and HEV infections.

## Acknowledgments

The authors are grateful to the volunteer participants and their families. To the pollsters, samplers and supervisors of the field work nationwide. Special thanks to the Directors of the

Reference Laboratories of DISA, DIRESA, GERESA and DIRIS, and to those responsible for the area of Regional Epidemiology. We thank to the Coordinators at the Estrategia Sanitaria de Prevención y Control de Infecciones de Transmisión Sexual, VIH/SIDA y Hepatitis B, Ministry of Health, Lima, Peru. We thank also to the researchers and professionals at the National Hepatitis Reference Laboratory of INS-Peru, Centro Nacional de Salud Pública, INS, Peru (Manuel Terrazas, Lorena Santos, Helena and María Martínez), for the processing and obtaining of the samples for entry and coding in the Netlab system.

## Author Contributions

**Conceptualization:** Cesar Cabezas, Omar Trujillo, Ángel Gonzales-Vivanco, Carlos Manuel Benites Villafane, Johanna Balbuena, Alfredo Oswaldo Borda-Olivas, Magna Aurora Suarez-Jara, Flor de María Peceros.

**Data curation:** Cesar Cabezas, Max Carlos Ramírez-Soto.

**Formal analysis:** Cesar Cabezas, Omar Trujillo, Ángel Gonzales-Vivanco, Carlos Manuel Benites Villafane, Johanna Balbuena, Magna Aurora Suarez-Jara, Flor de María Peceros.

**Funding acquisition:** Cesar Cabezas, Omar Trujillo.

**Investigation:** Cesar Cabezas, Omar Trujillo, Ángel Gonzales-Vivanco, Carlos Manuel Benites Villafane, Johanna Balbuena, Alfredo Oswaldo Borda-Olivas, Magna Aurora Suarez-Jara, Flor de María Peceros, Max Carlos Ramírez-Soto.

**Methodology:** Cesar Cabezas, Omar Trujillo, Johanna Balbuena, Alfredo Oswaldo Borda-Olivas, Magna Aurora Suarez-Jara, Flor de María Peceros, Max Carlos Ramírez-Soto.

**Project administration:** Omar Trujillo.

**Resources:** Max Carlos Ramírez-Soto.

**Software:** Max Carlos Ramírez-Soto.

**Supervision:** Cesar Cabezas, Omar Trujillo, Ángel Gonzales-Vivanco, Johanna Balbuena, Magna Aurora Suarez-Jara, Flor de María Peceros.

**Validation:** Alfredo Oswaldo Borda-Olivas.

**Writing – original draft:** Cesar Cabezas, Omar Trujillo, Ángel Gonzales-Vivanco, Carlos Manuel Benites Villafane, Johanna Balbuena, Alfredo Oswaldo Borda-Olivas, Magna Aurora Suarez-Jara, Flor de María Peceros, Max Carlos Ramírez-Soto.

**Writing – review & editing:** Cesar Cabezas, Max Carlos Ramírez-Soto.

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
