## [Decision Letter · Decision Letter 0]

8 Apr 2020

PONE-D-20-04808

Seroepidemiology of hepatitis A, B, C, D and E virus infections in the general population of Peru: a cross-sectional study

PLOS ONE

Dear Dr. Ramírez-Soto,

Thank you for submitting your manuscript to PLOS ONE. After careful consideration, we feel that it has merit but does not fully meet PLOS ONE’s publication criteria as it currently stands. Therefore, we invite you to submit a revised version of the manuscript that addresses the points raised during the review process by the first reviewer.

We would appreciate receiving your revised manuscript by May 23 2020 11:59PM. To enhance the reproducibility of your results, we recommend that if applicable you deposit your laboratory protocols in protocols.io, where a protocol can be assigned its own identifier (DOI) such that it can be cited independently in the future. For instructions see: http://journals.plos.org/plosone/s/submission-guidelines#loc-laboratory-protocols

We look forward to receiving your revised manuscript.

Kind regards,

Pierre Roques, Ph.D.

Academic Editor

PLOS ONE

Journal Requirements:

2)  We note that you have indicated that data from this study are available upon request. PLOS only allows data to be available upon request if there are legal or ethical restrictions on sharing data publicly. For information on unacceptable data access restrictions, please see http://journals.plos.org/plosone/s/data-availability#loc-unacceptable-data-access-restrictions.

Reviewers' comments:

Reviewer's Responses to Questions

**Comments to the Author**

1. Is the manuscript technically sound, and do the data support the conclusions?

Reviewer #1: Yes

Reviewer #2: Yes

2. Has the statistical analysis been performed appropriately and rigorously? 

Reviewer #1: N/A

Reviewer #2: Yes

3. Have the authors made all data underlying the findings in their manuscript fully available?

Reviewer #1: No

Reviewer #2: Yes

4. Is the manuscript presented in an intelligible fashion and written in standard English?

Reviewer #1: Yes

Reviewer #2: Yes

5. Review Comments to the Author

Reviewer #1: This manuscript describes, the seroepidemiology of hepatitis A, B, C, D and E virus infections in the general population of Peru, in a cross sectional study. The focus was on the participants of both sexes, aged 15–69 years, who had lived for >6 months in a specific region of Peru. In a total of 5183 study participants, the prevalence rates of anti-HAV, HBsAg, total anti-HBc, anti-HBs, anti-HCV, anti-HDV and anti-HEV were 98.4%, 0.38%, 10.25%, 59.9%, 0.12%, 1.77% and 14% respectively. On this basis, the authors conclude that the prevalence of HAV and HEV in the population aged 15–69 years in Peru is high, while the prevalence of HBV and HDV has changed from intermediate to low endemicity level and the prevalence of HCV is low.

Although the research question and sample collection is quite interesting, I have some minor and major concerns:

Minor comments:

Lines 29-30: The authors must specify the anti-HAV and anti-HEV antibodies are anti-HEV and anti-HAV IgG.

Lines 161-163: In my opinion, this sentence should be corrected, because it is rather antibodies HBe and not anti-HBc IgM.

Lines 163-164: Why do the authors report only the characteristics (sensitivity and specificity) of the HEV IgG ELISA kit, and they did give not characteristics of the other tests or kits (HBV, HCV, HAV, and HDV).

Lines 153-157: Why do the authors tested only the IgG against HEV and HAV which indicate only past infection or vaccination, rather than acute infection.

How do the authors manage to explain the presence of total anti-HBc and the absence of anti-HBc IgM?

Lines173-174: If the tests used for the analysis of Anti HBs markers are both quantitative and qualitative tests, I think the authors should specify it for a better understanding.

Line 192: The authors must specify the type of immunoglobulin

Line 254: The authors must specify the type of immunoglobulin

Lines 303-305: The authors report that the low prevalence of HBsAg, and total anti-HBc testify to level low endemicity. What conclusion will the authors draw, if the total 10% anti-HBc were anti-HBc IgM?

Lines 351-354: What do the authors mean by incidence and rate?

Lines 377-378: The authors use which study to state that exposure to hepatitis virus infections is similar in both sexes

Major comments:

Lines 126-128: Does not the proportional distribution across each region in the country according to the respective regional population size, bias the data, given that the authors themselves report to the existence of areas of high and low prevalence of infection.

Lines 170-171: If anti-HBc IgM and anti-HBs are positive, this can it be defined as a past infection?

Lines 211-214: The authors report that all participants reactive to HBsAg reagent were chronic carriers of HBV (reactive to anti-HBe reagent). However, only data of total anti-HBc antibody are available (table 1). The data on HBeAg, anti-HBe, and levels of anti- HBs are absent.

What do the authors mean by reactive to anti-HBe reagent? Were the antibodies anti-HBe negative or positive?

Lines 268-271: The authors cannot say that this is co-infection, because the markers tested show only old contact with HAV and HEV.

Lines 275-277: Data on anti-HBs levels are absent.

Lines 277-280: The authors cannot make a comparison with this study where the children were between 0-15 years old, since in their study, the children were between 15-18 years old.

Lines 287-289: Are these variations not rather associated with the different age groups targeted in these studies?

Lines 294-300: The authors must moderate their arguments, because the drinking water and sewerage services being available for the entire population of the metropolitan area of Lima, the high prevalence observed cannot be justified solely by the migration.

Lines 316-317: The authors mean that all HBs positive samples had a level > 10 mIU / ml. If yes, the authors must specify it. If not, the authors must give the methodology used to obtain the 59%, because the data on the levels of anti HBs is not available.

Reviewer #2: Well designed study with good results that show the importance of Population based vaccination campaigns and the impact of improving other public health measures in Hepatitis prevention.

Appreciate the good background but would ask for a few recent epidermiological references to be included ( M Alavi et al in BMC infectious diseases and M Jeffries et al in World Journal of clinical cases) to bring some current literature into the article.

Very good discussion and well set.

6. PLOS authors have the option to publish the peer review history of their article (what does this mean?). If published, this will include your full peer review and any attached files.

Reviewer #1: No

Reviewer #2: No

---

## [Author Response · Author response to Decision Letter 0]

19 May 2020

Response-to-reviewers: Manuscript PONE-D-20-04808

We thank the Reviewers for their comments and constructive criticism, we believe that the quality of our manuscript has been significantly improved. We have revised our paper in a point-by-point manner. Modifications are in yellow text. 

Reviewer #1: This manuscript describes, the seroepidemiology of hepatitis A, B, C, D and E virus infections in the general population of Peru, in a cross sectional study. The focus was on the participants of both sexes, aged 15–69 years, who had lived for >6 months in a specific region of Peru. In a total of 5183 study participants, the prevalence rates of anti-HAV, HBsAg, total anti-HBc, anti-HBs, anti-HCV, anti-HDV and anti-HEV were 98.4%, 0.38%, 10.25%, 59.9%, 0.12%, 1.77% and 14% respectively. On this basis, the authors conclude that the prevalence of HAV and HEV in the population aged 15–69 years in Peru is high, while the prevalence of HBV and HDV has changed from intermediate to low endemicity level and the prevalence of HCV is low.

Although the research question and sample collection is quite interesting, I have some minor and major concerns:

Minor concerns

1. Lines 29-30: The authors must specify the anti-HAV and anti-HEV antibodies are anti-HEV and anti-HAV IgG.

Response: Thank you for this suggestion; we have corrected the text (lines 29-30).

Lines 161-163: In my opinion, this sentence should be corrected, because it is rather antibodies HBe and not anti-HBc IgM.

Response: Thank you for your comment; There is no mistake the sentence is correct (lines 161-163). Anti-HAV IgG, anti-HCV, anti-HEV IgG and anti-HBs were analyzed by enzyme-linked immunosorbent assay (ELISA).

Lines 163-164: Why do the authors report only the characteristics (sensitivity and specificity) of the HEV IgG ELISA kit, and they did give not characteristics of the other tests or kits (HBV, HCV, HAV, and HDV).

Response: Thank you for this suggestion; we have corrected the text (lines 159-166). The sensitivity of HBsAg is 99.88% and the specificity is 99.75%, the sensitivity of anti-HBc IgG is 99.92% and the specificity is 99.82%. Anti-HAV IgG, anti-HCV, anti-HEV IgG and anti-HBs were analyzed by enzyme-linked immunosorbent assay (ELISA) (Beijing Wantai Biological Pharmacy, Beijing, China). Of note, the sensitivity of anti-HAV IgG is 100% and the specificity is 99%, the sensitivity of anti-HCV is 99.88% and the specificity is 99.87%, the sensitivity of anti-HEV IgG is 99% and the specificity is 99.99%, and the sensitivity of anti-HBs is 100% and the specificity is 99.58%.

Lines 153-157: Why do the authors tested only the IgG against HEV and HAV which indicate only past infection or vaccination, rather than acute infection.

Response: Thank you for your comment. Our aim was to determine the prevalence of HAV and HEV infections in asymptomatic carriers. Due to this we do not test IgM for HAV and HEV. 

How do the authors manage to explain the presence of total anti-HBc and the absence of anti-HBc IgM?

Response: Thank you for your comment. Past infection was defined as the presence of total anti-HBc IgG and the absence of HBsAg.

Lines173-174: If the tests used for the analysis of Anti HBs markers are both quantitative and qualitative tests, I think the authors should specify it for a better understanding.

Response: Thank you for this suggestion; we have corrected the text (lines 177-179).

Line 192: The authors must specify the type of immunoglobulin

Response: Thank you for this suggestion; we have corrected the text (line 197)

Lines 303-305: The authors report that the low prevalence of HBsAg, and total anti-HBc testify to level low endemicity. What conclusion will the authors draw, if the total 10% anti-HBc were anti-HBc IgM?

Response: There would be cases of acute infection. However, in our study all HBsAg-positive participants were chronic inactive carriers of HBV (anti-HBe positive, and HBeAg and anti-HBc IgM negatives) (lines 216-217). 

Lines 351-354: What do the authors mean by incidence and rate?

Response: Thank you for your comment; we have corrected the text (lines 357-360). 

Lines 377-378: The authors use which study to state that exposure to hepatitis virus infections is similar in both sexes

Response: Thank you for your comment; we have corrected the text. This was corrected "However, our findings should not be affected by this, since prevalence to hepatitis HAV, HBV and HEV infections was similar in both sexes" (lines 382-383).

Major comments:

Lines 126-128: Does not the proportional distribution across each region in the country according to the respective regional population size, bias the data, given that the authors themselves report to the existence of areas of high and low prevalence of infection.

Response: Thank you for your comment. Despite to the existence of areas of high and low prevalence of HBV and HDV, our results showed a change in the level of endemicity. Therefore, these findings are the result of vaccination against HBV.

Lines 170-171: If anti-HBc IgM and anti-HBs are positive, this can it be defined as a past infection?

Response: Thank you for your comment, we have corrected the text. Past infection was defined as the presence of total anti-HBc IgG and the absence of HBsAg and anti-HBc IgM (lines 175-176). 

Lines 211-214: The authors report that all participants reactive to HBsAg reagent were chronic carriers of HBV (reactive to anti-HBe reagent). However, only data of total anti-HBc antibody are available (table 1). The data on HBeAg, anti-HBe, and levels of anti- HBs are absent.

Response: Thank you for your comment, we have corrected the text. “All HBsAg-positive participants were chronic inactive carriers of HBV (anti-HBe positive, and HBeAg and anti-HBc IgM negatives)”. In this study, the reactive HBsAg samples were analyzed by ELISA for HBeAg, anti-HBc IgM, antibodies against HBV envelope antigen (anti-HBe), anti-HDV IgM and anti-HDV IgG. Levels of anti-HBs area in table 1.

What do the authors mean by reactive to anti-HBe reagent? Were the antibodies anti-HBe negative or positive?

Response: Thank you for your comment, we have corrected the text. “All HBsAg-positive participants were chronic inactive carriers of HBV (anti-HBe positive, and HBeAg and anti-HBc IgM negatives)” (lines 216-217).

Lines 268-271: The authors cannot say that this is co-infection, because the markers tested show only old contact with HAV and HEV.

Response: Thank you for your comment, we have removed the text. 

Lines 275-277: Data on anti-HBs levels are absent.

Response: Levels of anti-HBs are in table 1.

Lines 277-280: The authors cannot make a comparison with this study where the children were between 0-15 years old, since in their study, the children were between 15-18 years old.

Response: Thank you for your comment, we have corrected the text. (lines 283-285)

Lines 287-289: Are these variations not rather associated with the different age groups targeted in these studies?

Response: Thank you for your comment, we have corrected the text. These variations in prevalence rates of anti-HAV IgG across different studies are probably related to differing hygiene habits, age groups studied, educational levels and local water and sanitation conditions. (lines 293-295)

Lines 294-300: The authors must moderate their arguments, because the drinking water and sewerage services being available for the entire population of the metropolitan area of Lima, the high prevalence observed cannot be justified solely by the migration.

Response: Thank you for your comment, we have corrected the text. Such discrepancy can be explained by the constant migration of populations, particularly in the age groups studied, from the interior of the country to Lima, hygiene habits, educational levels. We must also consider that, unlike in the metropolitan area of Lima, there are locations in the peri-urban areas of Lima that lack adequate water and sanitation facilities (lines 302-306).

Lines 316-317: The authors mean that all HBs positive samples had a level > 10 mIU / ml. If yes, the authors must specify it. If not, the authors must give the methodology used to obtain the 59%, because the data on the levels of anti HBs is not available.

Response: Thank you for your comment, we have corrected the text. It is also important to highlight that only 60% of the population in our study had seroprotective levels of anti-HBs �10 mUI/ml (lines 322-323)

Reviewer #2: Well designed study with good results that show the importance of Population based vaccination campaigns and the impact of improving other public health measures in Hepatitis prevention.

Appreciate the good background but would ask for a few recent epidermiological references to be included (M Alavi et al in BMC infectious diseases and M Jeffries et al in World Journal of clinical cases) to bring some current literature into the article.

Response: Thank you for your comment. The study of M Alavi et al. determined the mortality trends among people with hepatitis B and C in New South Wales, Australia. Due this we have not included this study because our aim was to determine the prevalence of HAV, HBV, HCV, HDV and HEV infections, as well as the seroprotective levels of anti-HBs, in the Peruvian population. The study of M Jeffries et al in World Journal of clinical cases was not identified.

---

## [Editor Report · Decision Letter 1]

22 May 2020

Seroepidemiology of hepatitis A, B, C, D and E virus infections in the general population of Peru: a cross-sectional study

PONE-D-20-04808R1

Dear Dr. Ramírez-Soto,

We are pleased to inform you that your manuscript has been judged scientifically suitable for publication and will be formally accepted for publication once it complies with all outstanding technical requirements.

With kind regards,

Pierre Roques, Ph.D.

Academic Editor

PLOS ONE
---

## [Editor Report · Acceptance letter]

1 Jun 2020

PONE-D-20-04808R1 

Seroepidemiology of hepatitis A, B, C, D and E virus infections in the general population of Peru: a cross-sectional study 

Dear Dr. Ramírez-Soto:

I am pleased to inform you that your manuscript has been deemed suitable for publication in PLOS ONE. Congratulations! Your manuscript is now with our production department. 

With kind regards,

on behalf of

Dr. Pierre Roques 

Academic Editor

PLOS ONE